# Convergent validity of video-based observer rating of drowsiness, against subjective, behavioral, and physiological measures

Yuji Uchiyama[1,4]*, Shunichiroh Sawai[2], Takuhiro Omi[2], Koichiro Yamauchi[2], Kimimasa Tamura[3], Takuya Sakata[2], Kiyofumi Nakajima[4], Hiroyuki Sakai[1]

1 Toyota Central R&D Labs., Inc., Nagakute, Aichi, Japan, 2 Woven Core, Inc., Chuo, Tokyo, Japan, 3 Toyota Research Institute Inc., Cambridge, MA, United States of America, 4 Toyota Motor Corporation, Toyota, Aichi, Japan

* uchiyama@mosk.tytlabs.co.jp

**Data Availability Statement:** All relevant data are included in the manuscript and its Supporting information files.

## Abstract

Driver drowsiness is a widely recognized cause of motor vehicle accidents. Therefore, a reduction in drowsy driving crashes is required. Many studies evaluating the crash risk of drowsy driving and developing drowsiness detection systems, have used observer rating of drowsiness (ORD) as a reference standard (i.e. ground truth) of drowsiness. ORD is a method of human raters evaluating the levels of driver drowsiness, by visually observing a driver. Despite the widespread use of ORD, concerns remain regarding its convergent validity, which is supported by the relationship between ORD and other drowsiness measures. The objective of the present study was to validate video-based ORD, by examining correlations between ORD levels and other drowsiness measures. Seventeen participants performed eight sessions of a simulated driving task, verbally responding to Karolinska sleepiness scale (KSS), while infra-red face video, lateral position of the participant's car, eye closure, electrooculography (EOG), and electroencephalography (EEG) were recorded. Three experienced raters evaluated the ORD levels by observing facial videos. The results showed significant positive correlations between the ORD levels and all other drowsiness measures (i.e., KSS, standard deviation of the lateral position of the car, percentage of time occupied by slow eye movement calculated from EOG, EEG alpha power, and EEG theta power). The results support the convergent validity of video-based ORD as a measure of driver drowsiness. This suggests that ORD might be suitable as a ground truth for drowsiness.

## Introduction

Driver drowsiness is a widely recognized cause of motor vehicle accidents. According to crash statistics, driver drowsiness accounted for 16% of all crashes in southwestern England in the United Kingdom that required police involvement [1], and 2.4% of vehicle crashes in the United States [2]. This wide range of the crash rates could be caused by the difference of

**Funding:** The authors received no specific funding for this work.

**Competing interests:** I have read the journal's policy, and the authors of this manuscript have the following competing interests: YU and HS are employed by Toyota Central R&D Labs., Inc. YU, and HS held patents related to slow eye movement detection. SS, TO, KY, KT, and TS are employed by Woven Core Inc. SS and TO have been members of Phase - 6 (FY2016 - 2020), Study Group for the Promotion of Advanced Safety Vehicles by the Ministry of Land, Infrastructure, Transport, and Tourism of Japan, and the Working Groups of "Technical Requirement and Issues of Emergency Driving Stop System" and "Practical Driver Monitoring Techniques." KN is employed by Toyota Motor Corporation. These do not alter our adherence to the PLOS ONE policy on sharing data and materials. However, the data that can specifically identify the research participants, such as face videos, are subject to restrictions on sharing as per the participants' informed consent.

criterion for identifying drowsiness-related crashes among the countries. In addition, case-control studies of public road crashes have revealed that, driver drowsiness increases vehicle crash risk [3–5]. A meta-analysis that included 70,098 drivers also showed that, driver drowsiness doubles the vehicle crash risk [6]. Thus, prevention of vehicle crashes induced by driver drowsiness is required.

The observer rating of drowsiness (ORD) has been used as a basis for driver drowsiness in many studies, investigating the crash risk of drowsy driving and drowsiness detection systems. The common procedure of ORD rating involves experienced human raters evaluating the drowsiness levels, based on visual observation of the driver [7–9]. The major advantage of ORD is that, driver drowsiness can be evaluated without driver distraction, as compared to probing methods such as the Karolinska sleepiness scale (KSS) [10], in which, drivers are verbally questioned about subjective sleepiness during driving. Therefore, several naturalistic driving studies have used ORD as the ground truth of driver drowsiness [11–13]. Hanowski et al. compared the ORD levels before critical incidents between at-fault and innocent haul drivers [11]. The results showed that the ORD levels of the at-fault drivers were significantly higher, than those of innocent drivers, this associating ORD with vehicle crash risk. Furthermore, another line of study to develop driver-drowsiness detection systems employed ORD [14–23]. Naurois et al. investigated neural networks with physiological, behavioral, and car measures, to predict driver drowsiness by using ORD [18]. Arefnezhad et al. showed that, deep neural networks with car measures can classify driver drowsiness using ORD [21].

Despite the widespread use of ORD, concerns remain about its convergent validity [24] in the literature till date. This validity is supported by the correlations between ORD and other drowsiness measures. Currently, several studies have shown correlations between ORD levels and other drowsiness measures. Wierwille et al. proposed a video-based ORD with 100-point continuous scale, using recorded facial video [7]. In their study, eight participants alternately performed a math task and a letter search task, while electroencephalogram (EEG), electrooculogram (EOG), and facial videos were recorded. Three trained raters evaluated ORD levels. The study briefly indicated non-negligible correlations between ORD levels and rate of eye closures, subjective drowsiness, task performance (i.e., math task and letter search task), and electrophysiological measures (i.e., EEG and EOG). However, the details of this study have not been fully described. Kitajima et al. proposed another ORD using a 5-point scale [8]. In their study, 12 participants performed 10 sessions of simulated driving tasks for 10 minutes, while facial videos were recorded. After each task, the participants answered a questionnaire (i.e., the Roken Mental Work Strain Checklist [25]), which included a subjective drowsiness scale. Two raters observed the video separated by 5 seconds, and evaluated the ORD level. The study only indicated correlation coefficients between ORD and subjective drowsiness levels for each participant, but did not provide other correlations. Anund et al. [9] proposed an ORD named B-ORS (behavioral observer rated sleepiness) with a 3-point scale. In their experiment, an ORD rater sitting in the backseat, evaluated the ORD level of twenty-four participants who drove an instrumented car on a public road. The results demonstrated that ORD was significantly correlated with KSS, blink duration, and vehicle lateral position. However, real-time rating might compromise the reliability of the results, as the rater referred to the KSS levels associated with the participants' verbal responses.

In the present study, to investigate the convergent validity of video-based ORD, we hypothesized that ORD is positively correlated with KSS, standard deviation of the lateral position (SDLP) [26, 27], percentage of time occupied by eye closure (PERCLOS) [28], percentage of time occupied by slow eye movement (SEM), EEG alpha power, and EEG theta power. The percentage of time occupied by SEM, EEG alpha, and theta power was used to validate the KSS, and was positively correlated with KSS [10, 29]. PERCLOS is positively correlated with

KSS during psychomotor vigilance tasks [30]. SDLP has been improved by the administration of wake-promoting drugs [31, 32]. Administration of modafinil improved SDLP in patients with narcolepsy or hypersomnia, compared to the placebo condition [31]. Administration of methylphenidate also improved SDLP in healthy volunteers, compared to the same condition [32]. Modafinil and methylphenidate block dopamine reuptake [33]. This is considered to maintain the activity of dopamine neurons in the reticular activating system, which controls the arousal level.

The objective of this study was to investigate the convergent validity of video-based ORD as a measure of driver drowsiness. Participants performed simulated driving tasks while the face videos, KSS, lateral position of the participant's car, eye closure, EOG, and EEG were recorded. Three experienced raters evaluated ORD levels using facial video observations. To examine the validity of the ORD, correlation coefficients of all combinations of drowsiness measures of ORD, KSS, SDLP, PERCLOS, percentage of time occupied by SEM, EEG alpha power, and EEG theta power were statistically tested.

## Materials and methods

### Participants

A total of 24 participants, aged between 20 and 29 years, were recruited by a subject recruitment service (OMRON EXPERTLINK Co. Ltd., Kyoto, Japan) from January to March 2021, and data from 17 of them were analyzed. All participants included in the study met the specified inclusion and exclusion criteria as follows. To avoid bias in both sleepiness and driving performance, the participants had to meet the following criteria: a score of less than 15 on the Center for Epidemiological Studies-Depression Scale (CES-D) [34], an intermediate type of morningness–eveningness questionnaire (MEQ) [35, 36], not having had any experience of shift work in the three months before the experiment, no travel to different time zones during the three months before the experiment, not using drugs more than double the administration interval, non-smoker, a body mass index (BMI) of lower than 25, and a score of less than or equal to 21 on the short motion sickness susceptibility questionnaire (MSSQ-Short) [37] (to exclude the top 20% population who was susceptible to motion sickness). Additionally, participants had to possess a valid car driving license. The exclusion criterion was wearing glasses to measure eye movement. However, participants wearing contact lenses were eligible to participate.

To select participants with changing drowsiness levels during the experiment, a criterion of KSS range was set. The criterion was that the number of KSS responses with $KSS \geqq 8$ as well as with $KSS \leqq 7$, was greater than two.

The Ethics Committee of Toyota Motor Corporation (Toyota, Aichi, Japan) provisionally approved the experiment protocol (experiment approval number, 2020QV0048). The protocol was revised according to the comments from the provisional approval and was fully approved. Written informed consent was obtained from all participants. This study was conducted in accordance with the Declaration of Helsinki. The authors had access to information that could identify individual participants (i.e., full names, and face videos) during or after data collection.

### Apparatus

A steering wheel, accelerator, brake pedals (G29, Logitech, Lausanne, Switzerland), and seat were fixed on a frame (RCZ01, Costick, Nara, Japan). Driving simulator software (UC-win/road Version 10, Forum8, Tokyo, Japan) presented a simulated driving scene on a liquid crystal display (LCD-E425, NEC, Tokyo, Japan) placed in front of the frame. The mean distance

between the display and eyes of the participants was 91.1 cm (SD 7.5). The driving simulator sound and the synthesized instruction voice that requested verbal KSS responses, were presented on an in-ear headphone. Driving data were recorded at 100-ms interval. Three infrared cameras of an eye-movement measuring device (Smart Eye, Smart Eye AB, Gothenburg, Sweden) were placed around the steering column. Eye movement data were recorded at a 60-Hz sample rate. An infrared camera (driver monitoring system, Aisin, Aichi, Japan) [20], to record the video of a driver's face for rating the ORD, was placed on the steering column. Participants' faces were recorded from the lower frontal position using the camera. The size of the recorded face was approximately 1/4–1/3 of the width of the image. The camera systems for the eye movements measurement and the driver's face video recording mounted infra-red lights synchronized to capturing video frames, respectively. Because of the difference of infra-red light wavelength between the camera systems, the systems did not interfere each other. Active electrodes connected to a bio-amplifier (Polymate V AP5148, Miyuki Giken Co., Ltd., Tokyo, Japan), were attached to the participants for EOG and EEG measurements.

## Procedure

The data collection for this study was conducted from January to March 2021 in a quiet meeting room at a hotel near Shin-Toyota station, located in Toyota City, Japan. The participants arrived at the room at 9, received explanations of the experiment, and practiced the experimental tasks. After practice, electrodes for physiological measurements were attached to the skin. The participants then performed three experimental sessions (S1–S3) before the one-hour lunch time, and five sessions (S4–S8) after lunch time. Eight sessions were conducted between about 10:00 and 16:30. In one session, participants performed an 18-min simulated driving task, after performing a 10-min psychomotor vigilance task (PVT) [38, 39] and filled out a Japanese version of the KSS [29]. During the session, the participant sat on the seat of the DS, and eye movements, EOG, EEG, and videos were recorded. To induce drowsiness in the participants, the instructions of the experiment were displayed on the LCD monitor, and the experimenters rarely spoke to the participants, during the sessions. The illuminance around the driver's seat is approximately 100 lx. The room temperature was approximately 22˚C–26˚C. PVT data were used for the other analyses.

## Simulated driving task

The participants drove a car in the left lane of a four-lane monotonous expressway at night (Fig 1A). The participants drove steadily within the lane, keeping a steady distance from the car ahead, by controlling the steering wheel and the accelerator and brake pedals. The car ahead moved at a speed of 95 km/h. While driving, the participants verbally reported their sleepiness on the KSS, when the voice of "KSS" was presented. The relationship between KSS levels and the description of drowsiness (S2 Table) was shown in the bottom left corner of the display, so that the driver could refer to it while driving. There were no other cars, except for the participant's car and the car ahead of it.

The road structure for the driving task consisted of three repetitions of two types of successive road segments, during a session (Fig 1B). The first road segment consisted of five repetitions of the set of gentle left and right curves, to evaluate SDLP and physiological measures. The road segment length was such that the car could travel for five minutes when it travelled at the same speed as the car ahead (i.e., 95 km/h). Following the first segment, the second road segment was a straight road that aided verbal responses to the KSS. When the participant's car passed at the starting position of the straight road segment, the synthesized voice of "KSS" was presented, and the participants verbally responded to the KSS level. The length of the second

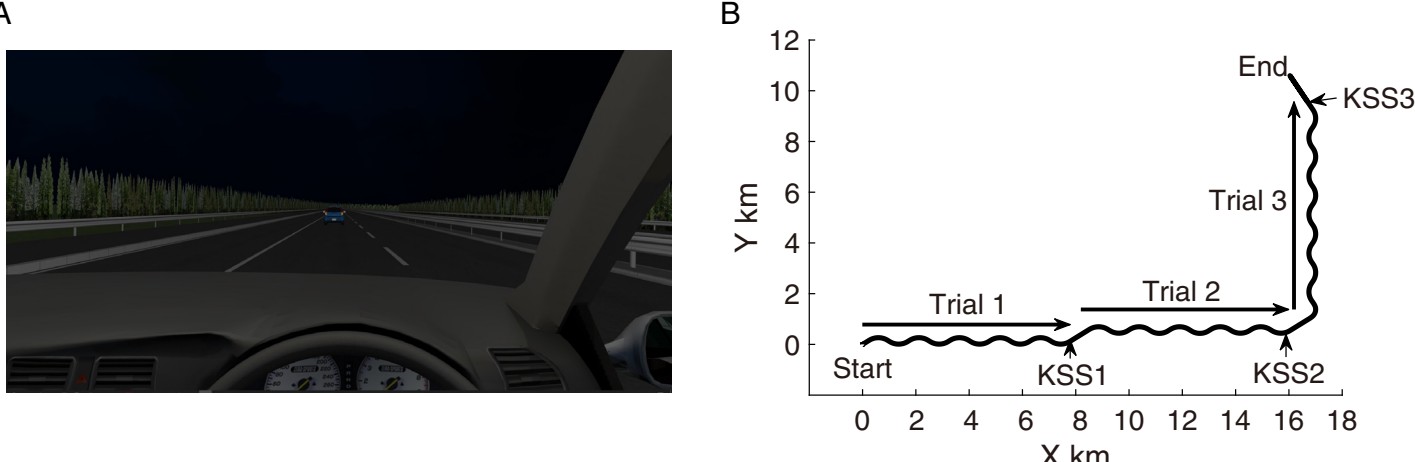

**Fig 1. Frontal view and road geometry of the driving simulator.** A: Participants drove a car on a left lane of a monotonous expressway at night, following a leading car driven at 95 km/h. B: The black thick line indicates the road structure for one driving session. The participant's car ran from Start to End. The arrow lines with Trial1 -3 indicate the road segments for SDLP measurement, and physiological measures (i.e., PERCLOS, SEM, and EEG). The length of the segment is suitable for driving the car for 5-minutes, with a maintained speed of 95 km/h. KSS1—3 indicate the positions of presenting a synthesized voice of "KSS," which requests the response of the KSS to the participant.

road segment was such that, the car could travel for one minute when it travelled at the same speed as the car ahead. These three repetitions of the road geometry corresponded to three 5-minute task trials, in one simulated driving task. The trial started from 5-minutes before the position presented "KSS", and ended to the position presented "KSS".

### Video-based observer rating of drowsiness (ORD)

The ORD in the present study followed the guidelines for driver monitoring systems of the Ministry of Land, Infrastructure, Transport, and Tourism of Japan [40]. The ORD levels in the guidelines are D1–D5 and S, with increasing drowsiness from D1 to D5 (Table 1). In the present study, the ORD levels from D1 to D5 were the ORD score from one to five points. The ORD level of "S" corresponded to five points. The guidelines recommend that raters subjectively evaluate the ORD level with the observation of face videos, so that the intervals of D1–D5 is equal. However, these guidelines do not include a detailed procedure to determine ORD levels. Thus, Woven Core, Inc. developed the ORD rating procedure based on the study by Kitajima et al. [8].

Three raters subjectively evaluated the ORD level for each 5-second video segment, according to the drowsiness descriptions (Table 1), considering the typical behavior for each ORD level (Table 1). The raters prioritized subjective ratings, noting that the ORD levels of D1 to D5 were at equal intervals. Thus, a video segment assigned to a certain ORD level does not necessarily have to include typical behaviors at that level. Raters were prohibited from consulting each other. The ORD score of the 5-second video segment was the mean ORD score across the three raters.

Before rating the experiment video, inter-rater reliability was assessed. The three raters, who had conducted ORD ratings prior to this study, rated three different 30-minute videos prepared for the assessment of their reliability. The concordance rate was calculated for all combinations of the three raters, according to the formula for "Concordance rate" in 5.2.2. of

**Table 1. Typical behaviors for observer rating of drowsiness.**

| ORD level (points) | Drowsiness description | Typical observed behavior |
|---|---|---|
| D1(1) | Not drowsy | Fast and frequent eye movements |
| | | Frequent body movements |
| | | Eye blinks with fast eyelid movements |
| | | Regular time intervals between eye blinks |
| | | Active body movements |
| D2(2) | Slightly drowsy | Slow saccadic eye movements |
| | | Lips open |
| | | Infrequent eye movements |
| | | Drooping eyelids |
| D3(3) | Moderately drowsy | Frequent slow eye blinks |
| | | Mouth movement |
| | | Sitting position change |
| | | Touching face |
| | | Frequent eye blinks |
| | | Less than half obscured pupils |
| | | Yawning |
| | | Tired-complexion |
| D4(4) | Very drowsy | Voluntary eye blinks |
| | | Frequent yawning and deep breathing |
| | | Unnecessary body movements such as shaking head and up and down movement of the shoulders, etc. |
| | | Slow blinking and SEM |
| | | Staring blankly at a single point |
| | | Inability to focus |
| | | More than half obscured pupils |
| D5(5) | Extremely drowsy | Closed eyes |
| | | Forward tilted head |
| | | Backward tilted head |
| | | Sagging cheeks |
| S(5) | Sleeping | D5 continues and does not awaken. The state is judged as "likely to be asleep". |

ORD—Observer Rating of Drowsiness; SEM—Slow Eye Movement; Notes: To determine the ORD level, the typically observed behaviors are helpful, but not always observed for an assigned ORD level. Japanese version is attached in S1 Table.

European Union (EU) law [41]. The concordance rate was greater than or equal to 0.7 for all combinations of raters.

After confirming the inter-rater reliability, the three raters evaluated the ORD level with 5-second video segments cut out from each 5-minute trial in the present study. The raters were not biased by other sleepiness measures. The ORD score for each trial was the average of the 5-second ORD score across all three raters, and the 5-minute trial.

## Karolinska sleepiness scale during the driving task

The KSS of the English version of EU law, for approval of driver drowsiness and attention warning (DDAW) [41], was translated into Japanese for the present study. Two researchers

discussed and translated sleepiness descriptions of KSS (S2 Table), with reference to the Japanese version [29] translated from the original pencil-and-paper KSS [10]. The participants maintained a sleep diary during their bedtime and rising hours, for four days prior to the experiment, recording KSS level, time of awakening, and time of onset of sleep, to train the KSS rating.

## Physiological measures

EOG and EEG were recorded at a sampling rate of 1 kHz, using a 60-Hz notch filter. The ground electrode of bio-amplifier, was attached to the middle sagittal plane of the forehead. Analyses were performed using MATLAB (R2021a, MathWorks, Inc., MA, US).

**Slow eye movements (SEM).** The percentage of time occupied by SEM in a trial, was derived from the EOG signal. The EOG electrodes were attached approximately 1 cm in the lateral direction from the lateral canthus of the left and right eyes. Differences in EOG signals between the left and right EOG electrodes, were applied with a low-pass filter of 15 Hz using EEG processing software (AP Viewer, Miyuki Giken Co., Ltd., Tokyo, Japan). To use an automatic SEM detection algorithm [42], EOG calibration data were recorded before sessions S1 and S4, for each participant. This data was measured while the participant moved their eyes to gaze at the three horizontal fixation points located at the center, left and right of the display. The distance between the display and the center of the participant's eyes, was also measured. The proportionality constants from the EOG voltage to the visual angle, were calculated from calibration data, those from the calibration data before S1, were applied to the EOG of sessions S1–S3, and those before S4 were applied to the EOG of S4–S8. Visual angle EOG data were applied to the SEM detection algorithm. The algorithm classified EOG data as SEM, using the following two criterion: (1) the eye movement speed was slower than 30°/s and (2) the movement distance satisfying the previous speed condition, was longer than 5° [42]. Time taken by SEM in a trial was calculated as the percentage of time occupied by SEM.

**EEG alpha and theta power.** The EEG alpha and theta power were calculated for each trial and each participant. The EEG voltage signal at C3 according to the 10–20 system with a right earlobe reference, was processed using a high-pass filter with a time constant of 0.3 second and a 30-Hz low-pass filter using the EEG processing software (AP Viewer, Miyuki Giken Co., Ltd., Tokyo, Japan). The filtered EEG signal was separated into a 15-second EEG segment, without artifacts, during the trials. The spectral power density in each of the remaining segments was estimated using the Welch method, with the "pwelch" function in the Signal Processing Toolbox of MATLAB. Spectral power of theta band (4–7.9 Hz [29]) and alpha band (8–12 Hz [29]), averaged across all valid EEG segments of the trial, was calculated from spectral power density, using the "bandpower" function of MATLAB.

## Percentage of eyelid closure over time (PERCLOS)

PERCLOS was determined from the pupil diameters of the eye movement data, according to the method of Chua et al. [30]. Eye closure was judged to have occurred, when the pupil diameter was lower than and equal to 20% of its maximum size, for more than 400 ms. The percentage of eye closure time in a trial was calculated using PERCLOS. The maximum pupil diameter was the mode of frequency of the pupil diameter during the PVT task, with the lowest KSS value for each participant.

## Statistical analysis

To ascertain the changes in drowsiness during the experiment, one-way repeated analysis of variance (ANOVA) was applied to ORD, KSS, SDLP, percentage of time occupied by SEM,

EEG alpha, and theta power, with trial as the independent variable in the "ranova" function in the Statistics and Machine Learning Toolbox of MATLAB. The statistical significance of these tests was set at $P < 0.05$.

To investigate the convergent validity of the ORD, correlation coefficients were calculated as follows: Pearson's correlation coefficients between all combinations of drowsiness measures (i.e., mean ORD, KSS, SDLP, PERCLOS, percent of time occupied by SEM, mean EEG alpha power, and mean EEG theta power) were calculated across trials for each participant using the "corr" function in the Statistics and Machine Learning Toolbox of MATLAB. Fisher's z-transformation was applied to the correlation coefficient for each participant. The transformed correlation coefficient was tested with a one-sample t-test, to test the null hypothesis that the mean correlation coefficient across the participants will be 0. This was conducted through the"ttest" function in the Statistics and Machine Learning Toolbox of MATLAB. Statistical significance was set at $P < 0.05$.

## Results

### Participants

We analyzed the data of 17 participants [7 women and 10 men; mean age 23 (SD 2.8) years; range 20–29 years], who satisfied the criterion of KSS range. The mean MEQ score was 52.0 (SD 4.6), mean CES-D score was 9.6 (SD 2.1), mean BMI was 20.8 (SD 1.8), mean MSSQ-Short score was 4.5 (SD 5.0), mean sleep duration before the experiment was 5.9 hours (SD 0.73), and mean wake up time was 6:24 (hh:mm), calculated from the sleep diaries. The demographic information for each participant is provided in the S1 File.

### Time course of drowsiness measures

It was ascertained that all drowsiness measures changed across the trials. The ANOVA indicated significant trial effects of ORD ($F(23, 368) = 9.75$, $P < 0.001$) (Fig 2A), KSS ($F(23, 368) = 10.86$, $P < 0.001$) (Fig 2B), SDLP ($F(23, 368) = 4.91$, $P < 0.001$) (Fig 2C), PERCLOS ($F(23, 368) = 4.82$, $P < 0.001$) (Fig 2D), percentage of time occupied by SEM($F(23, 368) = 3.79$, $P < 0.001$) (Fig 2E), EEG alpha power ($F(23, 368) = 4.80$, $P < 0.001$) (Fig 2F), and EEG theta power ($F(23, 368) = 2.10$, $P = 0.002 < 0.05$) (Fig 2G).

### Convergent validity of ORD

Correlations between drowsiness measures were examined to investigate the validity of the ORD. The relations between the ORD score and the other drowsiness measures are plotted in Fig 3. Increases in KSS, SDLP, PERCLOS, and the percentage of time occupied by SEM with ORD were observed in the plots.

Some participants became drowsy while driving. Therefore, some trials with a very large SDLP (maximum SDLP was approximately 400 cm), are shown in Fig 3B.

Relation of increasing EEG alpha power and EEG theta power, with increasing ORD, remained unclear. This lack of clarity depended on the large difference in EEG power between participants. For example, the larger EEG power of Subj05, compared to the other participants in Fig 3E and 3F, clearly indicates an increase.

Subsequently, the correlation coefficients between all combinations of drowsiness measures were examined, and presented a positive and statistically significantly different result, from 0. The correlation coefficients and their statistical test results were shown in Table 2.

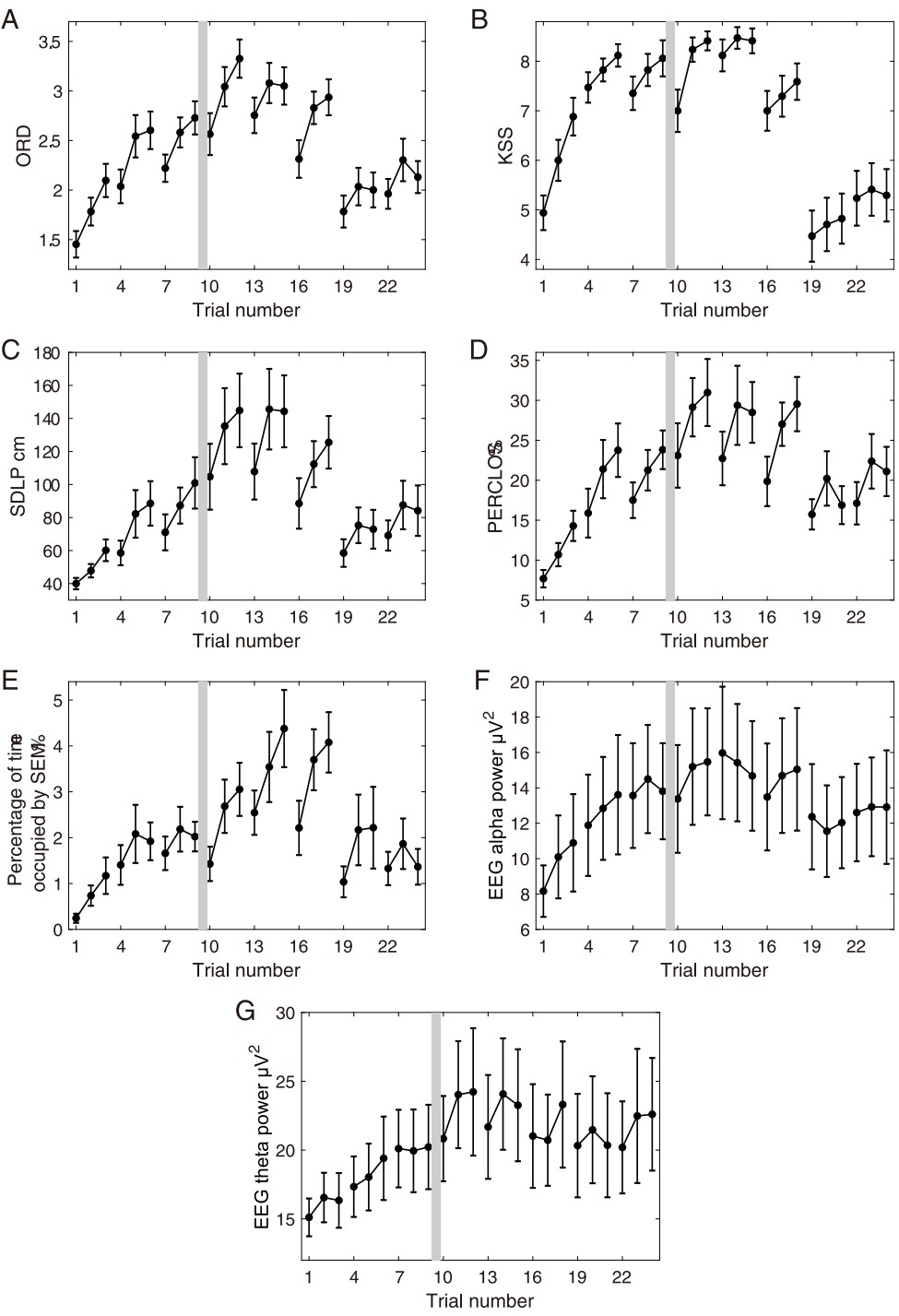

**Fig 2. Time course of drowsiness measures.** Each drowsiness measure averaged across 17 participants, was plotted against trial order. The drowsiness measures are, A: observer ratings of drowsiness (ORD), B: Karolinska sleepiness scale (KSS), C: standard deviation of lateral position (SDLP), D: percentage of eye closure (PERCLOS), E: the percentage of time occupied by slow eye movements (SEM), F: electroencephalographic (EEG) alpha power, and G: EEG theta power. Error bars indicate standard error. Lines connecting markers indicate the trials in one session. Gray area in each plot indicates lunch time.

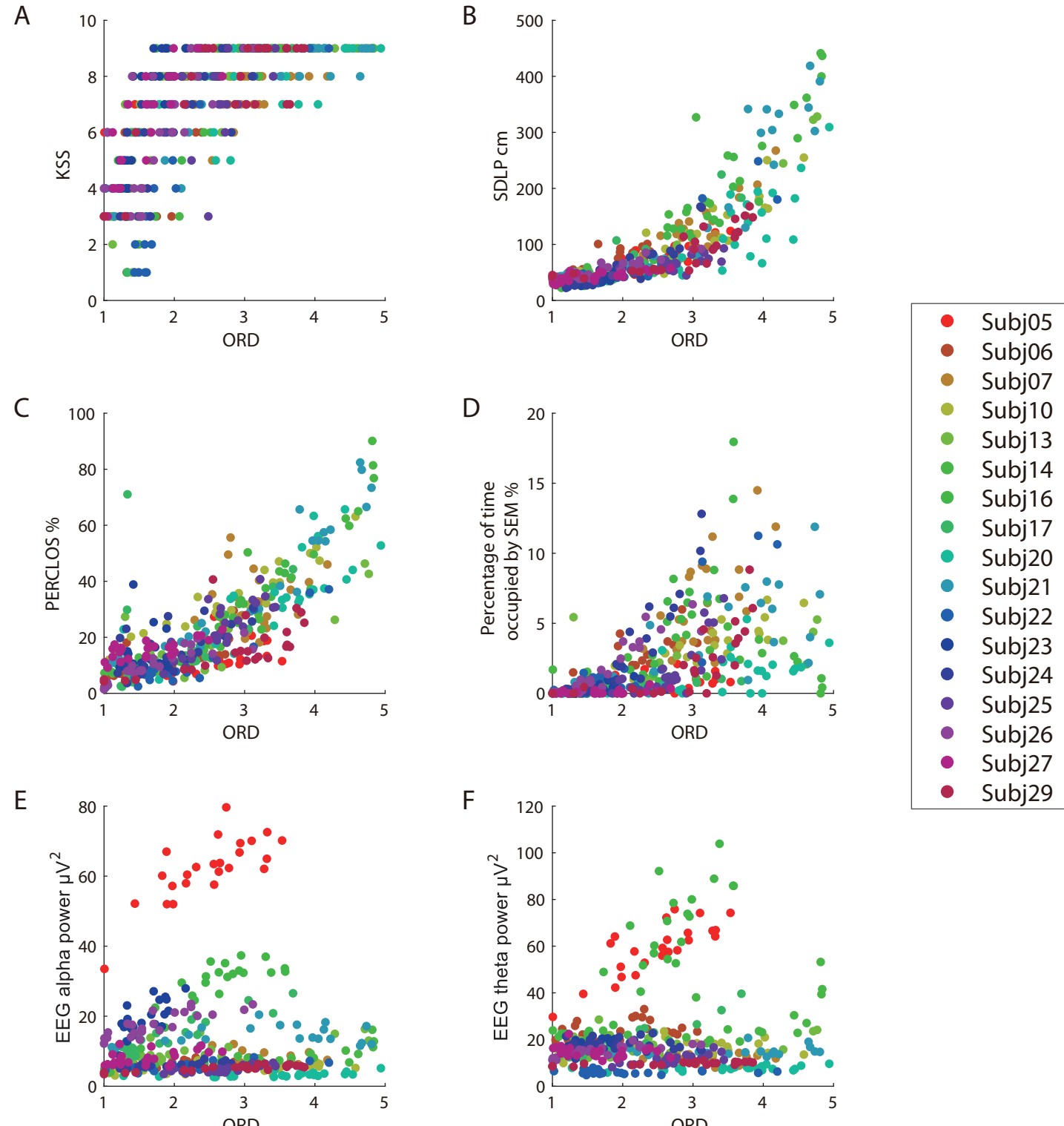

**Fig 3. Relations between the ORD and other drowsiness measures.** A: KSS, B: SDLP, C: PERCLOS, D: Percentage of time occupied by SEM, E: EEG alpha power and F: EEG theta power. Each marker indicates the data of one trial. Each color indicates a participant.

**Table 2. Correlation coefficients after Fisher's z-transformation between drowsiness measures.**

| Variables | 1 | 2 | 3 | 4 | 5 | 6 |
|---|---|---|---|---|---|---|
| 1. ORD | | | | | | |
| 2. KSS | 1.00*** (0.88–1.11) | | | | | |
| 3. SDLP | 1.29*** (1.15–1.43) | 0.71*** (0.60–0.82) | | | | |
| 4. PERCLOS | 1.06*** (0.78–1.34) | 0.55*** (0.39–0.72) | 1.07*** (0.70–1.43) | | | |
| 5. Percentage of time occupied by SEM | 0.98*** (0.75–1.20) | 0.60*** (0.45–0.75) | 1.13*** (0.81–1.44) | 0.80*** (0.45–1.15) | | |
| 6. EEG alpha power | 0.88*** (0.67–1.09) | 0.62*** (0.41–0.84) | 0.73*** (0.58–0.89) | 0.61*** (0.43–0.80) | 0.55*** (0.39–0.71) | |
| 7. EEG theta power | 0.52** (0.24–0.81) | 0.24* (0.05–0.43) | 0.42** (0.18–0.66) | 0.45*** (0.23–0.66) | 0.30* (0.07–0.53) | 0.66*** (0.37–0.95) |

ORD, Observer Rating of Drowsiness; KSS, Karolinska Sleepiness Scale; SDLP, Standard Deviation of Lateral Position,; PERCLOS, PERcentage of eye CLOSure; SEM, Slow Eye Movement; EEG, electroencepharogram; The values are mean Fisher's z-values transformed from Pearson correlation coefficients across 17 participants. The z-values were tested with one-sample t-test. Parentheses indicate 95% confidence interval.

*, $P < 0.05$;

**, $P < 0.01$;

***, $P < 0.001$

## Discussion

The convergent validity of the ORD was investigated by examining the correlations between the drowsiness measures. Participants performed a simulated driving task, while the drowsiness measures were recorded. The results indicated that ORD was significantly and positively correlated with KSS, SDLP, PERCLOS, percentage of time occupied by SEM, EEG alpha power, and EEG theta power (Table 2). Furthermore, the correlations between all combinations of other drowsiness measures, were significant and positive. These positive correlations support the convergent validity of ORD, as a measure of driver drowsiness.

The correlation between ORD and KSS in the present study, was similar results of the past studies which indicated the correlations between ORD and subjective drowsiness. Anund et al. [9] showed that ORD (i.e., B-ORS) was positively and significantly correlated with KSS which was verbally responded during driving. Kitajima et al. [8] showed a positive correlation trend between ORD on which the ORD and a subjective sleepiness (i.e., the Roken Mental Work Strain Checklist [25]). Wierwille et al. [7] showed the high positive correlation coefficient between their ORD and subjective drowsiness rated with an adjustable bar-knob control. ORD and subjective drowsiness in these three studies are not completely the same as those of the present study. However, all of the results in the past and the present results shows a relation between drowsiness rated by human raters and subjective drowsiness.

The positive correlations of ORD with PERCLOS, EEG alpha power, and EEG theta power in the present study, was supported by past studies of drowsiness. Wirewille et al. [7] indicated positive correlation coefficient of ORD with PERCLOS, EEG alpha power, and EEG theta power. Past studies showed that PERCLOS is a drowsiness measure. PERCLOS was significantly correlated with PVT lapses [28, 30]. Past studies also showed the relation between EEG power and drowsiness measures other than ORD. Meta analysis of EEG studies which compared EEG power between alert and drowsy conditions showed that EEG alpha and theta

power in drowsy condition were higher than those in alert condition [43]. EEG alpha and theta power correlated with PVT lapses [30] and also with KSS [10, 29].

The correlation between the ORD and SDLP in the present study, was similar to that reported by Anund et al. [9]. Their ORD was positively, but not significantly, correlated with SDLP, but this study portrayed a significantly positive correlation. This difference can be attributed to the structure of the road. Arien et al. showed that the SDLP on curved roads was larger than that on straight roads [44]. Anund et al. conducted their experiment on actual public roads, resulting in the inclusion of the effects of both road geometry and driver drowsiness, in the SDLP. In contrast, the present study evaluated the SDLP on almost similarly structured road for each trial, to exclude the road geometry effect. This difference may have decreased the noise of the SDLP and increased the statistical power in this experiment.

The correlation between the ORD and percentage of time occupied by SEM in the present study further supports the validity of the ORD. Previous validation studies [7–9] did not show a correlation between ORD and the percentage of time occupied by SEM. SEM can be considered an established indicator of drowsiness. SEM was observed just before behaviorally defined sleep onset [45] and is used in polysomnography as an indicator of sleep onset [46]. The percentage of time occupied by SEM in a drowsy state is associated with KSS [10], simple reaction time with a foot [47] and number of crashes in driving simulators [42].

The car dynamics model in the DS used in the present study is suitable for SDLP measurement. Previous studies that used the same DS showed that, alcohol consumption and sleep deprivation affected the SDLP [48, 49]. This means that SDLP evaluated by the DS in the present study is sensitive towards detecting driver impairment.

Correlations among all combinations of drowsiness measures in the present study, also support the validity of ORD. As the drowsiness measures selected here showed an increasing trend with increased drowsiness, positive correlations were expected between all combinations of such measures. As expected., the correlation coefficients for all combinations of drowsiness measures, returned a significantly positive outcome.

The convergent validity of the ORD in the current study, bridges the gap between ORD studies and other drowsiness studies that do not use ORD as a drowsiness measure. For example, validity raises the possibility of effectively preventing drowsiness, which has been studied based on other drowsiness measures. Reyner et al. indicated that caffeine was effective in preventing drowsy driving, evaluated with KSS, the number of incidents of simulated driving, and EEG power, in which they did not use the ORD [50]. If the ORD does not represent drowsiness, it is not clear whether caffeine intake in response to a warning from a driver drowsiness detection system developed based on the ORD, can effectively prevent drowsy driving. Our study bridges this gap, by suggesting the effectiveness of caffeine intake in such situations.

The rating procedure of ORD in the present study differs in some respect from other ORDs proposed in past studies [7–9]. The criteria for evaluating drowsiness levels, the numbers allotted for drowsiness levels, and the numbers of raters differed among the ORDs. Although all ORDs have in common that the human raters visually evaluate the driver for drowsiness, it should be noted that these difference may affect the results of ORD studies.

A limitation of the present study is that participants were youths (i.e., in their twenties). Visual components, an important requirement for rating ORD, alter with age. Palpebral fissure and peak eyelid velocity during spontaneous blinks, decreases with age [51]. The effect of age-related changes in visual factors, on the ratings of the ORD, remain elusive, and requires further future investigations.

Further research is required to establish ORD as a useful ground truth. The main reasons ORD is not widely adopted are its time-consuming nature and labor-intensive nature. To address these challenges, ORD rating should be automated with artificial intelligence that can

learn the association between ORD score rated by human raters and features extracted from facial videos. We believe that such an approach could greatly improve the usefulness of ORD as a ground truth for measuring sleepiness and enable its application in a broader range of settings.

## Conclusion

The participants performed a simulated driving task, while drowsiness measures were recorded. Three raters evaluated the ORD level, by observing the facial videos of those participating in the task. ORD was significantly and positively correlated with KSS, SDLP, PERCLOS, percentage of time occupied by SEM, EEG alpha power, and EEG theta power. The results support the convergent validity of video-based ORD, as a measure of driver drowsiness.

## Supporting information

**S1 Table. Japanese version of the typical behavior towards ORD.** Table 1 was translated from this Japanese version.
(PDF)

**S2 Table. Japanese version of the Karolinska sleepiness scale (KSS).** This KSS was translated from the English version of the scale in EU Law, to approve driver drowsiness and attention warning (DDAW) [41].
(PDF)

**S1 File. Demographic data for each participant.** The Excel file contains subject ID, sex, age, MEQ score, CES-D score, BMI, MSSQ-Short score, sleep duration before the experiment, and wake up time, for 17 participants.
(XLSX)

**S2 File. Trial data for Figs 2 and 3, and Table 2.** The Excel file contains subject ID, session ID, trial ID, mean ORD, KSS, SDLP, PERCLOS, percentage of time occupied by SEM, mean EEG alpha power, and mean EEG theta power, for 408 trials (i.e., three trials in eight sessions of 17 participants each).
(XLSX)

## Acknowledgments

The authors thank Naoto Yamada, and Tetsuhiro Itoh of Toyota Technical Development Corporation for assistance with data collection.

## Author Contributions

**Conceptualization:** Yuji Uchiyama, Shunichiroh Sawai, Takuhiro Omi, Koichiro Yamauchi, Kimimasa Tamura, Takuya Sakata, Kiyofumi Nakajima.

**Data curation:** Yuji Uchiyama, Shunichiroh Sawai, Takuhiro Omi, Koichiro Yamauchi, Kimimasa Tamura, Takuya Sakata.

**Formal analysis:** Yuji Uchiyama.

**Investigation:** Yuji Uchiyama.

**Methodology:** Yuji Uchiyama.

**Project administration:** Yuji Uchiyama, Kiyofumi Nakajima.

**Resources:** Kiyofumi Nakajima.

**Software:** Yuji Uchiyama.

**Supervision:** Shunichiroh Sawai, Kiyofumi Nakajima.

**Visualization:** Yuji Uchiyama.

**Writing – original draft:** Yuji Uchiyama, Hiroyuki Sakai.

**Writing – review & editing:** Yuji Uchiyama, Shunichiroh Sawai, Takuhiro Omi, Koichiro Yamauchi, Kimimasa Tamura, Takuya Sakata, Kiyofumi Nakajima, Hiroyuki Sakai.

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
