## [Decision Letter · Decision Letter 0]

20 Mar 2023

PONE-D-22-27691Convergent validity of video-based observer rating of drowsiness, against subjective, behavioral, and physiological measuresPLOS ONE

Dear Dr. Uchiyama,

Thank you for submitting your manuscript to PLOS ONE. After careful consideration, we feel that it has merit but does not fully meet PLOS ONE’s publication criteria as it currently stands. Therefore, we invite you to submit a revised version of the manuscript that addresses the points raised during the review process.

Please consider all comments Please submit your revised manuscript by 28 March 2023. If you will need more time than this to complete your revisions, please reply to this message or contact the journal office at plosone@plos.org. Please include the following items when submitting your revised manuscript:A rebuttal letter that responds to each point raised by the academic editor and reviewer(s). You should upload this letter as a separate file labeled 'Response to Reviewers'.A marked-up copy of your manuscript that highlights changes made to the original version. You should upload this as a separate file labeled 'Revised Manuscript with Track Changes'.An unmarked version of your revised paper without tracked changes. You should upload this as a separate file labeled 'Manuscript'.If applicable, we recommend that you deposit your laboratory protocols in protocols.io to enhance the reproducibility of your results. Protocols.io assigns your protocol its own identifier (DOI) so that it can be cited independently in the future. For instructions see: https://journals.plos.org/plosone/s/submission-guidelines#loc-laboratory-protocols. Additionally, PLOS ONE offers an option for publishing peer-reviewed Lab Protocol articles, which describe protocols hosted on protocols.io. Read more information on sharing protocols at https://plos.org/protocols?utm_medium=editorial-email&utm_source=authorletters&utm_campaign=protocols.

We look forward to receiving your revised manuscript.

Kind regards,

Ahmed Mancy Mosa, Ph.D.

Academic Editor

PLOS ONE

2. Please update the ethics statement to indicate that the considerations included in the provisional accept issued biy the Ethics Committee were addressed and full approval was obtained. Please note that further consideration of this manuscript will require this consideration to be addressed.

“Toyota Central R&D Labs., Inc. (https://www.tytlabs.com/), Woven Core, Inc. (https://www.woven-planet.global/en/woven-core), and Toyota Motor Corporation, (https://global.toyota/en/) supported the funding and salaries of authors YU, SS, TO, KY, KT, TS, KN, and HS, but did not have any additional role in the study design, data collection and analysis, decision to publish, or preparation of the manuscript. The specific roles of these authors are articulated in the “author contributions” section.”

“I have read the journal’s policy, and the authors of this manuscript have the following competing interests: YU and HS are employed by Toyota Central R&D Labs., Inc. YU and HS held patents related to slow eye movement detection. SS, TO, KY, KT, and TS are employed by Woven Core Inc. SS and TO had been members of Phase - 6 (FY2016 - 2020), Study Group for the Promotion of Advanced Safety Vehicle by Ministry of Land, Infrastructure, Transport and Tourism of Japan and Working Group of “Technical requirement and issues of emergency driving stop system” and for “Practical driver monitoring techniques”. KN is employed by Toyota Motor Corporation. These do not alter our adherence to PLOS ONE policies on sharing data and materials, except for those that include the privacy and personal information of research participants.”

Reviewers' comments:

Reviewer's Responses to Questions

**Comments to the Author**

1. Is the manuscript technically sound, and do the data support the conclusions?

Reviewer #1: Yes

Reviewer #2: Yes

Reviewer #3: Yes

2. Has the statistical analysis been performed appropriately and rigorously? 

Reviewer #1: Yes

Reviewer #2: Yes

Reviewer #3: Yes

3. Have the authors made all data underlying the findings in their manuscript fully available?

Reviewer #1: Yes

Reviewer #2: Yes

Reviewer #3: Yes

4. Is the manuscript presented in an intelligible fashion and written in standard English?

Reviewer #1: Yes

Reviewer #2: Yes

Reviewer #3: Yes

5. Review Comments to the Author

Reviewer #1: The introduction clearly presents the problem(s) and the current state of knowledge as it relates to the topic and its significance. The methods sections and procedures are clear and reproducible given the needed resources. The analysis and accompanying tables are appropriate and clear. The discussion and limitations are clearly related to the real world issues related to the problem of fatal car accidents.

Reviewer #2: Comments

The manuscript is well written, but a couple of things need to be considered.

1. Why does the number of participants differ from what is mentioned in the abstract and in the methods section? In the abstract, you mentioned 17 participants, and in the methods, a total of 24 participants.

2. You included the inclusion criteria, but what were the exclusion criteria?

3. In line 126, the procedure section, mention the city name.

4. Consider rewriting the last sentence at line 375.

5. References: The references are not appropriate. And consider changing the reference numbers 1, 2, 3, and 6. Please erase reference number 11.

Reviewer #3: This is a well written paper. It is important to evaluate the Karolinska sleepiness scale in different countries or regions setting. I would suggest the authors to include some demographic description of the participants in supplementary materials.

6. PLOS authors have the option to publish the peer review history of their article (what does this mean?). If published, this will include your full peer review and any attached files.

Reviewer #1: No

Reviewer #2: No

Reviewer #3: No

---

## [Author Response · Author response to Decision Letter 0]

14 Apr 2023

Our responses to the reviewers are included in the attached file of Response_to_Reviewers.docx.

---

## [Decision Letter · Decision Letter 1]

26 Apr 2023

Convergent validity of video-based observer rating of drowsiness, against subjective, behavioral, and physiological measures

PONE-D-22-27691R1

Dear Dr. Uchiyama,

We’re pleased to inform you that your manuscript has been judged scientifically suitable for publication and will be formally accepted for publication once it meets all outstanding technical requirements.

Kind regards,

Ahmed Mancy Mosa, Ph.D.

Academic Editor

PLOS ONE

Additional Editor Comments (optional):

Reviewers' comments:

Reviewer's Responses to Questions

**Comments to the Author**

1. If the authors have adequately addressed your comments raised in a previous round of review and you feel that this manuscript is now acceptable for publication, you may indicate that here to bypass the “Comments to the Author” section, enter your conflict of interest statement in the “Confidential to Editor” section, and submit your "Accept" recommendation.

Reviewer #2: All comments have been addressed

2. Is the manuscript technically sound, and do the data support the conclusions?

Reviewer #2: Yes

3. Has the statistical analysis been performed appropriately and rigorously? 

Reviewer #2: Yes

4. Have the authors made all data underlying the findings in their manuscript fully available?

Reviewer #2: Yes

5. Is the manuscript presented in an intelligible fashion and written in standard English?

Reviewer #2: Yes

6. Review Comments to the Author

Reviewer #2: All comments have been addressed. I would request the editor to accept this manuscript for publication. Thank you.

7. PLOS authors have the option to publish the peer review history of their article (what does this mean?). If published, this will include your full peer review and any attached files.

Reviewer #2: No

---

## [Editor Report · Acceptance letter]

28 Apr 2023

PONE-D-22-27691R1 

Convergent validity of video-based observer rating of drowsiness, against subjective, behavioral, and physiological measures 

Dear Dr. Uchiyama:

I'm pleased to inform you that your manuscript has been deemed suitable for publication in PLOS ONE. Congratulations! Your manuscript is now with our production department. 

Kind regards, 

on behalf of

Dr. Ahmed Mancy Mosa 

Academic Editor

PLOS ONE